# k$^*$means: A Parameter-free Clustering Algorithm

## 1 Introduction

### Abstract

Clustering is a widely used and powerful machine learning technique, but its effectiveness is often limited by the need to specify the number of clusters, $k$, or by relying on thresholds that implicitly determine $k$. We introduce k$^*$means, a novel clustering algorithm that eliminates the need to set $k$ or any other parameters. Instead, it formulates the clustering problem as minimising a three-part encoding of the data. It uses this formulation to determine the optimal number of clusters, $k^*$, by splitting and merging clusters while also optimising the standard $k$-means objective. We prove that k$^*$means is guaranteed to converge and demonstrate experimentally that it significantly outperforms existing methods in scenarios where $k$ is unknown. We also show that it accurately estimates $k$ and that, empirically, its runtime is competitive with existing methods and scales well with dataset size.

## 2 Introduction

Clustering is a fundamental task in machine learning. As well as allowing data visualisation and exploration, it is used for several more specific functions in the context of machine learning systems, such as representation learning (Liu et al., 2023a; Niu et al., 2024), federated learning (Ma et al., 2023), exploration in reinforcement learning (Wagner & Harmeling, 2024), anomaly detection (Markovitz et al., 2020), and has found widespread application in the natural sciences (Xu et al., 2025; Kisi et al., 2025; Meyer et al., 2025; Hebdon et al., 2025). It has also been interwoven with deep learning feature extraction in the areas of deep clustering (Caron et al., 2018; Mahon & Lukasiewicz, 2021; Miklautz et al., 2024; Liu et al., 2023b; Vo et al., 2024) and deep graph clustering (Mo et al., 2024; Fini et al., 2023). Clustering can produce meaningful and interpretable partitions of data, even in the absence of information often required by other machine learning methods, such as annotated labels.

However, almost all existing clustering algorithms still require some user-set parameters, which limits their applicability to cases where the user can choose appropriate values. Two common classes of clustering algorithms are centroid-based and density-based. The former, typified by $k$-means, work by finding the optimal location for cluster centre-points (centroids), and then assigning points to nearby centres. These algorithms generally require the user to specify the number of clusters. Density-based algorithms aim to locate clusters where the density of points is high. They also require some threshold(s) to determine what constitutes a high-density region and where to separate them.

In this paper, we design a clustering algorithm that eliminates the need to set the number of clusters, tunable thresholds, or any other parameters. Our algorithm, k$^*$means, extends $k$-means by automatically determining the number of clusters, $k$, that minimises the length of a three-part coding of the data. The first part, which we refer to as the *model cost*, describes centroids of the fit clustering model, the second part, which we refer to as the *index cost*, describes the cluster assignments, and the third, which we refer to as the *residual cost*, describes the displacement of each point from its assigned centroid. Too many clusters incur prohibitively high index costs, while too few incur prohibitively high residual costs, so the objective guides the model towards a reasonable value of $k$.

This is a similar approach to the minimum description length (MDL) principle, which states that the optimal data representation is the one that uses the fewest bits. `k*means` differs from existing MDL clustering approaches (Kontkanen et al., 2005) in that it uses hard labels for each point, whereas MDL typically takes the description length as the negative log probability under a probabilistic model such as a Gaussian mixture model (GMM), and this amounts to labelling each point as a soft label of a distribution across all clusters.

We optimise our three-part code objective by including two subclusters per cluster in the model. The "assign" and "update" steps of $k$-means are applied to the subclusters in the same way as to the main clusters, and the algorithm has the option to split a cluster into its two subclusters or merge two clusters if it reduces the description length.

Despite its simplicity, $k$-means remains the most widely used clustering algorithm because it is fast, provably converges, has just one easily interpretable parameter, and achieves accuracy competitive with more complicated methods. We aim to maintain these advantages with `k*means`. We provide a proof that `k*means` is also guaranteed to converge. Additionally, our experiments show that `k*means` largely maintains the speed and accuracy advantages of $k$-means. It is as fast or faster than most other $k$-agnostic clustering methods, scales well with dataset size, and is close to or on par with the accuracy of $k$-means, even when $k$-means has an oracle for the true value of $k$. We also demonstrate, in synthetic experiments, that it can identify $k$ more accurately than existing methods. Our contributions are summarised as follows:

- We introduce `k*means`, an entirely parameter-free clustering algorithm;

- We give a formal proof that `k*means` will convergence in finite time;

- We design synthetic data experiments to test whether `k*means` can infer the true value of $k$, and show that it can with much higher accuracy than existing methods;

- We show experimentally that, with respect to standard clustering metrics, it is more accurate than all existing methods that do not require setting $k$ and is as fast as, or faster than, most of these methods.

The remainder of this paper is organised as follows. Section 3 discusses related work, Section 4 describes the algorithm of `k*means` in detail, Section 6 presents experimental results, and finally Section 7 concludes and summarises.

## 3 Related Work

Two well-known centroid-based clustering algorithms are $k$-means, (MacQueen, 1967; Lloyd, 1982) and GMMs (Dempster et al., 1977). The former partition data into $k$ clusters by iteratively assigning points to the nearest centroid and updating centroids until convergence, and the latter which fit a multivariate normal model via expectation maximization. A number of more complex clustering algorithms are also in widespread use.

Spectral Clustering (Ng et al., 2001) transforms data using eigenvectors of a similarity matrix before applying a clustering algorithm such as $k$-means. Mean Shift (Comaniciu & Meer, 2002) discovers clusters by iteratively shifting points toward areas of higher density until convergence. It does not require setting $k$, but does require a bandwidth parameter. Meanshift++ (Jang & Jiang, 2021) and Quickshift (Vedaldi & Soatto, 2008) are two variants that speed up the slow, quadratic time of each step of Meanshift. Affinity Propagation (Frey & Dueck, 2007) identifies exemplars among data points and forms clusters by exchanging messages between pairs of samples until convergence. Like mean shift, it does not require specifying the number of clusters $k$, but instead relies on a preference parameter and a damping factor. A common drawback of both mean shift and affinity propagation is their quadratic space complexity, which limits scalability. Divisive hierarchical clustering continues to bifurcate clusters with $k$-means, $k = 2$, until a stopping criterion.

Two other classic methods that estimate $k$ automatically are DPC (Rodriguez & Laio, 2014) and DPMM (Antoniak, 1974). DPC is a density-based method that seeks to assign centroids to high-density regions and far apart from each other. DPMM models the data as generated from a mixture of Dirichlet processes, and fits an approximation with a Bayesian estimator.

DBSCAN (Ester et al., 1996) identifies dense regions as clusters by grouping points with many neighbours, while marking sparse points as noise. OPTICS (Ankerst et al., 1999) extends DBSCAN by ordering points by reachability distance, allowing it to identify clusters with varying densities. HDBSCAN (Campello et al., 2013) further builds on DBSCAN by introducing a hierarchical clustering framework that extracts flat clusters based on stability. Although DBSCAN and its variants do not require specifying the number of clusters, they rely on other parameters—such as `eps` and `min-pts`, which specify the neighbourhood size and the number of points required to form a 'dense region'. OPTICS avoids setting `eps` by computing reachability distances over a range of values, but in its place introduces a steepness parameter to define cluster boundaries (where the reachability value decreases faster than this steepness). Tuning these parameters can yield a wide range of values for the number of predicted clusters (see Appendix A). Thus, without knowledge about the number of clusters or parameter values, DBSCAN and its derivatives can be difficult to apply effectively.

X-Means (Pelleg & Moore, 2000) extends $k$-means by automatically determining the optimal number of clusters using the Bayesian information criterion (BIC) (Schwarz, 1978). Our method is similar to X-Means in two respects: firstly, in that it selects $k$ using an agnostic criterion from probability/information theory, and secondly, in that it considers bifurcating each centroid as the means by which to explore different values of $k$. The SMEM algorithm, (Li & Li, 2009) employs the same idea but for a GMM and using MDL as the stopping criterion. However, there are some important differences between k*means and these two methods. k*means uses three-part code length as the criterion, whereas X-Means uses BIC and Li & Li (2009) uses MDL. Secondly, our method does not require the $max_K$ parameter. It can, in principle, return any value of $k$ (although this would have to be bounded by $n$). Thirdly, X-means, and Li & Li (2009) have to run the standard EM-training algorithm to convergence each time a new value of $k$ is explored. k*means, in contrast, returns the best model in one stage by splitting only when it reduces the code length and keeping a pre-initialised pair of sub-centroids for each cluster, which are updated one step at a time as $k$ is optimised. This means k*means only needs to run $k$-means like updates (combined with the split and merge operations) to convergence once. Ishioka (2000) uses a very similar method to X-means, keeping a stack of clusters during training, and sequentially running $k$-means with $k=2$ on each. Again, this is much less efficient than k*means, which does not need to run multiple models to convergence. Also similar is the algorithm by Ronen et al. (2022), which splits and merges stochastically during deep clustering. The $k$-splits algorithm (Mohammadi et al., 2022) is a recent algorithm that performs divisive hierarchical clustering until the inter-centroid distance exceeds a threshold. Clustering applications often deal with unknown $k$ by training many $k$-means models with varying values of $k$, and selecting that with the lowest BIC (Zhang & Li, 2014; Lancaster & Camarata, 2019; Salmanpour et al., 2022). Selecting by silhouette score, or the elbow method, is also a common approach (Alam, 2023). Our experiments (Section 6) find that this is not only much slower than k*means, as it requires running many models to convergence, but also less accurate, often severely overestimating $k$. A summary of the clustering algorithms discussed in this section and their parameters is presented in Table 1.

## 4 The k*means Algorithm

In the exposition and accompanying algorithms in this Section, we use the following notation: $X = \{x_1, \ldots, x_N\} \subset \mathbb{R}^d$ is the data to be clustered, $k$ is the number of clusters, $\mu \in \mathbb{R}^{k \times d}$ is the matrix of mean vectors, $C$ is the partition, $\mu_s \in \mathbb{R}^{k \times 2 \times d}$ is the tensor of sub-centroids, and $C_s$ is the length $k$ array of binary partitions of each cluster. Indexing notation follows Python-style.

### 4.1 Quantifying Description Length

In k*means, we quantify a bit cost for the various components of a clustering model and how they change over training. This allows k*means to directly minimise the description length in a single procedure that simultaneously finds the optimal number of clusters, $k^*$, and fits a $k$-means model with $k^*$ clusters.

The bitcost of a data point $x$ under a clustering model has two parts, the cost of specifying which cluster it belongs to, which we call the *index cost*, and the cost of specifying its displacement from that cluster's centroid, which we call the *residual cost*. The former requires selecting an element of $\{0, \ldots, k-1\}$, thus taking $\log k$ bits. The latter, the code length of $x$, given its cluster centroid $c_x$, is expressed as $-\log p(x|c_x)$, following the correspondence between prefix-free codes and probability distributions Cover & Thomas (2006), where we

Table 1: Common clustering algorithms and their required parameters

| Algorithm | Required Parameters |
|---|---|
| K-means | Number of clusters ($k$) |
| Gaussian Mixture Models (GMM) | Number of components ($k$); Covariance type |
| Spectral Clustering | Number of clusters ($k$); Affinity type |
| Mean Shift | Bandwidth parameter (kernel width) |
| Affinity Propagation | Preference parameter; Damping factor |
| DBSCAN | Neighborhood radius (eps); Minimum points (minpts) |
| HDBSCAN | Minimum cluster size; Minimum samples; Cluster selection eps |
| X-Means | Maximum number of clusters; Minimum number of clusters |
| Divisive Hierarchical Clustering | Stopping criterion |
| k*means | — |

interpret the clusters probabilistically (see equation 2 below). We also need to represent the cluster centroids, themselves, which we term the *model cost*. Each centroid takes $dm$ bits, where $d$ is the dimensionality and $m$ is the floating point precision. (The precision $m$ is chosen from the data as the smallest value that allows perfect representation.)

Thus, the total cost to be minimised by k*means, is the centroid cost for each of $k$ centroids, plus the index cost for each data point plus the residual cost for each data point. Formally, let $X$ be the data to be clustered, $\Pi(X)$ be the set of all partitions of $X$, and $|P|$ be the number of subsets in a partition. The optimal partition $P^*$ is

$$P^* = \operatorname*{arg\,min}_{P \in \Pi(X)} |P|dm + |X| \log |P| - \sum_{x \in X} \log p(x|c_x) \,, \tag{1}$$

where $c_x$ is the cluster that $x$ was assigned to.

We model the cluster distribution as a multivariate normal distribution with unit variance

$$p(x|c) = \frac{1}{(2\pi)^{d/2}} \exp\left(-\frac{1}{2}(x-c)^T(x-c)\right) \tag{2}$$

$$\iff -\log p(x|c) = \frac{d \log 2\pi + ||x-c||^2}{2} \,. \tag{3}$$

Therefore, equation 1 becomes

$$P^* = \operatorname*{arg\,min}_{P \in \Pi(X)} |P|dm + |X| \log |P| + \frac{1}{2} \sum_{S \in P} Q(S) \,, \tag{4}$$

where $Q$ computes the total sum of squares: $Q(X) = \sum_{x \in X}(x-c_x)^2$ and then $k^* = |P^*|$. (Full derivation is provided in Appendix B).

This assumption of Gaussian cluster distributions with unit variance is a limitation of k*means. The MDL philosophy is to restrict to a class of models that are easy to work with, even if it may not contain the true generative distribution, and our empirical results validate that this is an effective approach for the problem of determining $k$ in clustering. However, future work should develop the current approach further by exploring other classes of models.

The dependence of some terms in equation 4 on the data scale but not others means that the solution found by k*means is dependent on the data scale. In particular, multiplying all data points by a large number

---

**Algorithm 1** K*-means Algorithm

---

1: **procedure** K*-MEANS($X$)
2:     best_cost $\leftarrow \infty$
3:     unimproved_count $\leftarrow 0$
4:     $\mu \leftarrow \frac{1}{n} \sum_{i=1}^{n} x_i$                    ▷ where the $x_i$s are the constituents of $X$, i.e. $X = \{x_1, \ldots, x_n\} \subset \mathbb{R}^d$
5:     $C \leftarrow [X]$    ▷ array of sets of vectors in $\mathbb{R}^d$, the $i$th element is the set of points in the $i$th cluster, initialised, using Python style notation for an array with a single element, $X$, meaning all points in one cluster
6:     $\mu_s \leftarrow$ sub-centroids initialised using $k$++means
7:     $C_s \leftarrow [\{x \in X : \|x - \mu_{s_1}\| < \|x - \mu_{s_2}\|\}, \{x \in X : \|x - \mu_{s_2}\| < \|x - \mu_{s_1}\|\}]$
8:     **while** true **do**
9:         $\mu, C, \mu_s, C_s \leftarrow$ KMEANSSTEP($X, \mu, C, \mu_s, C_s$) ▷ One assign + update step for both main centroids/clusters and subcentroids/subclusters.
10:        $\mu, C, \mu_s, C_s, \text{did\_split} \leftarrow$ MAYBESPLIT($X, \mu, C, \mu_s, C_s$)
11:        **if** did_split **then**
12:            $\mu, C, \mu_s, C_s \leftarrow$ KMEANSSTEP($X, \mu, C, \mu_s, C_s$)
13:        **else**
14:            $\mu, C, \mu_s, C_s, \text{did\_merge} \leftarrow$ MAYBEMERGE($X, \mu, C, \mu_s, C_s$)
15:            **if** did_merge **then**
16:                $\mu, C, \mu_s, C_s \leftarrow$ KMEANSSTEP($X, \mu, C, \mu_s, C_s$)
17:        cost $\leftarrow$ MBITCOST($X, \mu, C$)
18:        **if** cost $<$ best_cost **then**
19:            best_cost $\leftarrow$ cost
20:            unimproved_count $\leftarrow 0$
21:        **else**
22:            unimproved_count $\leftarrow$ unimproved_count $+ 1$
23:        **if** unimproved_count $=$ patience **then**
24:            **break**
25:    **return** $\mu, C$

26: **procedure** BITCOST($X, \mu, C$)
27:     $d \leftarrow$ the dimensionality of $X$
28:     $floatprecision \leftarrow -\log$ of the minimum distance between any values in $X$
29:     $floatcost \leftarrow \frac{max(X) - min(X)}{floatprecision}$        ▷ The max and min are taken across all coordinates across all data points.
30:     $modelcost \leftarrow |C|d \times floatcost$
31:     $idxcost \leftarrow |X| \log(|C|)$
32:     $c \leftarrow$ the sum of the squared distances of every point in X from its assigned centroid
33:     $residualcost \leftarrow \frac{|X|d \log(2\pi) + c}{2}$
34:     **return** $modelcost + idxcost + residualcost$

---

would cause k$^*$means to find more clusters. This is because it is interpreting distance as an absolute, not merely a relative, measure of similarity. Roughtly speaking, this measure is what is used to decide whether two points should be placed in the same cluster.

## 4.2 Minimising Description Length

In this section, we describe the algorithm by which k$^*$means efficiently optimises equation 4. For a more formal exposition, see Algorithm 1. In all algorithm definitions we use Python-style indexing notation. The familiar Lloyd's algorithm for $k$-means alternates between two steps: `assign`, which assigns each point to its nearest centroid, and `update`, which updates the centroids of each cluster to the mean of all of its assigned points. As well as the centroids and clusters, k$^*$means keeps track of subcentroids and subclusters. Subclusters consist of a partition of each cluster into two, and subcentroids are the means of all points in each subcluster. These are updated during the `update` and `assign` steps in just the same way as the main clusters and centroids. Essentially, each cluster has a mini version of $k$-means happening inside it during training.

k$^*$means introduces two additional steps, `maybe-split` and `maybe-merge`, to the standard `assign-update` procedure. After the `assign` and `update` steps, the algorithm calls `maybe-split`, which uses the subclusters

and subcentroids to determine whether any cluster should be split. If no clusters are split, it proceeds with `maybe-merge`. In the case of a split, each constituent subcluster is promoted to a full cluster, and a new set of subclusters and subcentroids is initialised within each of them, following the k++-means initialisation method (Arthur & Vassilvitskii, 2006). If two clusters are merged, their subclusters are discarded, and the clusters themselves are demoted to become two subclusters inside a new cluster that is their union. `k*means` is initialised with just a single cluster containing all data points (and its two sub-clusters), and then cycles between `assign`, `update`, `maybe-split`, and `maybe-merge` until the assignments remain unchanged for a full cycle. (In practice, for speed, we terminate if the cost has improved by $< 2$ in the past five cycles. These are not core parameters of the algorithm, and can easily be omitted, in which the runtime is $\sim 30\%$ longer.) In this way, it simultaneously optimises $k$ *and* the standard $k$-means objective, with respect to equation 4.

**Maybe-Split Step** This method (Algorithm 2) checks whether each cluster should be split into two. A naive approach would involve computing equation 4 for the current parameters, and then again with the given cluster replaced by its two subclusters, splitting if the latter is smaller. However, we can perform a faster, equivalent check by simply measuring the difference in cost. If there are currently $k$ clusters, splitting would increase the index cost of each point by $\log(k+1) - \log(k) \approx 1/(k+1)$. It would also decrease the residual cost by $Q(S) - (Q(S_1) + Q(S_2))$, where $S$ is the original cluster and $S_1, S_2$ are its subclusters. To determine whether a split is beneficial, we compute $Q(S) - (Q(S_1) + Q(S_2))$ for every cluster. If any value exceeds $2N/(k+1)$, the cluster with the largest difference is split.

**Maybe-Merge Step** This method (Algorithm 3) checks whether a pair of clusters should be merged. To avoid the time taken to compare every pair, we compare only the closest pair of centroids. Analogously to `maybe-split`, the potential change from merging is $\frac{1}{2}(Q(S) - (Q(S_1) + Q(S_2))) - n/k$, where $S_1$ and $S_2$ are the two clusters with the closest centroids, and $S = S_1 \cup S_2$. If this value is positive, then $S_1$ and $S_2$ are merged and become the new subclusters inside the new cluster $S$.

---

**Algorithm 2** `Maybe-Split` Procedure

---

1: **procedure** MAYBESPLIT$(X, \mu, C, \mu_s, C_s)$
2:     best_costchange $\leftarrow$ BITCOST$(X, \mu, C)$
3:     split_at $\leftarrow -1$
4:     **for** $i \in \{0, \dots, |\mu|\}$ **do**
5:         $subc1, subc2 \leftarrow C_s[i]$
6:         $submu1, submu2 \leftarrow \mu_s[i]$
7:         $costchange = \sum_{x \in submu1} \|x - subc1\|^2 + \sum_{x \in submu2} \|x - subc2\|^2 - \sum_{x \in C[i]} \|x - \mu[i]\|^2 + |X|/(|\mu|+1)$
8:         **if** costchange $<$ best_costchange **then**
9:             best_costchange $\leftarrow$ costchange
10:            split_at $\leftarrow i$
11:    **if** best_costchange $< 0$ **then**
12:        $\mu \leftarrow$ SPLIT$(\mu, \mu_s, \text{split\_at})$
13:        $C \leftarrow$ SPLIT$(C, C_s, \text{split\_at})$
14:        $submu1', submu2' \leftarrow$ k++means on $C_s[i]$
15:        $\mu_s \leftarrow \mu_s[: \text{split\_at}] + (submu1', submu2') + \mu_s[\text{split\_at} :]$
16:    **return** $\mu, C, \mu_s, C_s, \text{split\_at} \geq 0$
17: **procedure** SPLIT$(A, A_s, \text{split\_at})$
18:     $A \leftarrow A[: \text{split\_at}] + A_s[\text{split\_at}] + A[\text{split\_at} :]$

---

---

**Algorithm 3** `Maybe-Merge` Procedure

---

1: **procedure** MAYBEMERGE($X, \mu, C, \mu_s, C_s$)
2:      $i_1, i_2 \leftarrow$ the indices of the closest pair of centroids
3:      $Z \leftarrow C[i_1] \cup C[i_2]$
4:      $m_{\mathrm{merged}} \leftarrow \frac{1}{|Z|} \sum_{x \in Z} x$
5:      $mainQ \leftarrow \sum_{z \in Z} (z - m_{\mathrm{merged}})^2$
6:      $subcQ \leftarrow \sum_{x \in C[i_1]} \|x - \mu[i_1]\|^2 + \sum_{x \in C[i_2]} \|x - \mu[i_2]\|^2$
7:      $costchange \leftarrow mainQ - subcQ - n/|\mu|$
8:      **if** costchange $< 0$ **then**
9:          $C \leftarrow C$ with $C[i_1]$ replaced with $Z$ and $C[i_2]$ removed
10:         $\mu \leftarrow \mu$ with $\mu[i_1]$ replaced with $m_{\mathrm{merged}}$ and $\mu[i_2]$ removed
11:     **return** $\mu, C$, costchange $> 0$

---

## 5 Theoretical Guarantees

### 5.1 Proof of Convergence

We now prove that `k*means` is guaranteed to converge in finite time. This is an extension of the proof of convergence for $k$-means, and uses the fact that all four of the steps at each cycle–`assign`, `update`, `maybe-split`, and `maybe-merge`–can only decrease the cost function in equation 4.

**Lemma 1.** *At each `assign` step (which is the same step as vanilla k-means), the three-part code length either decreases or remains the same, and it remains the same only if no points are reassigned.*

*Proof.* As defined in Section 4.1, the three parts to the cost are the model cost, consisting of the bits to specify the centroids of the fit clustering model, the index cost, consisting of the bits to specify the cluster membership of each point, and the residual cost, consisting of the bits corresponding to the displacement of each from its cluster centre. The former depends only on the number of points $n$ and the number of clusters $k$, and is unaffected by re-assignment. The latter is proportional to the sum of squared distances of each point to its assigned centre. By definition of reassignment, if a point is reassigned, then it is closer to its new centroid than its old centroid. Thus, every reassignment does not affect the first two summands of the cost and strictly reduces the third. □

**Lemma 2.** *At each `update` step (which is the same step as vanilla k-means), the bit cost either decreases or remains the same, and it remains the same only if no centroids are updated.*

*Proof.* As with the assign step, the update step does not change $k$ or $n$ and so does not affect the index cost. The latter can be written as the sum across clusters of the sum of all points in that cluster from the centroid. When the centroids are updated, they are set to the mean of all points in the cluster, which is the unique minimiser of the sum of squared distances. As well as a standard statistical fact, this can be seen by observing that the $SS(y) = \sum_{i=1}^{m} (y - x_i)^2$ is a U-shaped function of variable $y$, so achieves its global minimum when

$$SS'(y) = 0 \iff$$
$$2\sum_{i=1}^{m} y - x_i = 2mx - 2\sum_{i=1}^{m} x_i = 0 \iff$$
$$y = \frac{1}{m} \sum_{i=1}^{m} x_i \,.$$

This holds for the reassignment of each cluster and, by extension, for the entire reassignment step. □

**Theorem 3.** *The `k*means` algorithm is guaranteed to converge in finite time.*

*Proof.* By Lemmas 1 and 2, the bit cost strictly decreases at each step at which points are reassigned and centroids updated. The other two steps, `maybe-split` and `maybe-merge`, include explicit steps for which

the bit cost decreases before being performed, so they are also guaranteed to strictly reduce the bit cost. Together, this means the algorithm will never revisit an assignment during training. Moreover, there are a finite number of assignments, equal to the number of partitions of $n$ data points, which is given by the $n + 1$th Bell number (Graham et al., 1994), $B_{n+1}$. Therefore k*means cannot run for more than a finite number, namely $B_{n+1}$, steps. □

**Remark 4.** *This is an extension of the standard proof of convergence of k-means. Like for k-means, this proof establishes a theoretical worst-case run time which is exponential, but then, in practice, the algorithm converges quickly. It is known that k-means is NP-hard (Drineas et al., 2004), and that it can, in theory, run in exponential time even in 2 dimensions (Vattani, 2009). As* k*means *subsumes k-means, the same is true of it. However, also like k-means, the practical runtime is very good. Practical empirical runtimes are studied in detail in Section 6.*

In Appendix C we also prove a lower bound on the performance of k*means. We show that, assuming each cluster is a multivariate normal distribution with unit variance, and centroids are separated by a distance $\delta$, then all $k$ centroids will be within $\epsilon$ of their true values, with probability at least $p$, as long as

$$\delta \ > \ \sqrt{\frac{1 + \epsilon^2 - (\epsilon^2/2 - 1)e^{-\epsilon^2/2}}{1 - e^{-\epsilon^2/2} - \sqrt[k]{p}}} + \epsilon\,.$$

### 5.2 Complexity Analysis

The computational complexity of each iteration of k*means is $O(n)$, the same as that of $k$-means. The steps are as follows: (1) assign and update the clusters and centroids, (2) assign and update the subclusters and subcentroids, (3) check whether to split, (4) (if (3) returns true) split, (5) (if (3) returns false) check whether to merge, (6) (if (5) returns true) merge. Steps (1) and (2) are the usual Lloyd's algorithm steps, requiring the computation of the distance of each point from the centroids. Both are $O(n)$. Step (3) requires computing the distance of each point from its assigned centroid, and from its assigned subcentroid, so it is also $O(n)$. Splitting requires initialising new subcentroids with k++means, also $O(n)$. Step (5) requires computing, for the closest pair of centroids, the distance of each point in their union from their assigned centroid and from the mean of the union. Again, this is $O(n)$. Finally, step (6) requires redefining the merged centroid as the mean already computed in (5), and reindexing the existing clusters. This is $O(1)$. Thus, the total complexity is $5O(n) + O(1) = O(n)$.

## 6 Experimental Evaluation

We evaluate our clustering algorithm with three sets of experiments. Firstly, we use synthetic data, controlling the true number of clusters, and test whether the algorithm correctly identifies it. Secondly, we measure performance on labelled data, and compare the predicted cluster labels to the true class labels using supervised clustering metrics. Thirdly, we examine the runtime as a function of dataset size, and show that it scales well compared to existing methods.

### 6.1 Synthetic Data

For a range of values of $k$, we first use Bridson sampling (Bridson, 2007) to sample $k$ centroids in $\mathbb{R}^2$ near the origin with a minimum inter-point distance of $\delta$. Then we sample $1000/k$ points from a multivariate normal distribution, with unit variance, centred at each centroid. Examples of this synthetic data with varying $\delta$ are shown in Figure 1. We then run k*means and comparison methods on these 1,000 points and compare the number of clusters they find to $k$. We repeat this 10 times each for each $(k, \delta) \in \{1, \ldots, 50\} \times \{2, 3, 4, 5\}$. For each value of $\delta$, there are 10 examples each of 50 different values of $k$. We compute the accuracy, i.e., fraction of these 500 examples with perfectly correct prediction of $k$, and also the mean squared error (MSE) from the predicted $k$ to the true $k$.

Table 2 presents the results. As shown, k*means consistently outperforms the baseline algorithms in inferring $k$. Unsurprisingly, its performance improves as the distance between centroids increases, and notably, the accuracy

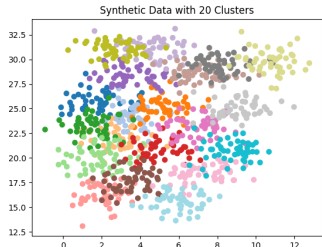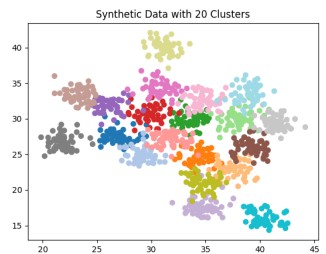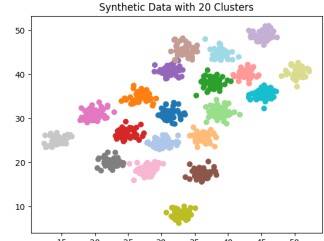

Figure 1: Synthetic data of standard, multivariate Normal clusters, with varying degrees of separation. Left: weak separation, inter-centroid distance constrained to $\geq 2$, k*means is 9% accurate in inferring $k$ and baselines are $\leq 4.4\%$. Middle: inter-centroid distance constrained to $\geq 3$, k*means is 25% accurate in inferring $k$ and baselines are $\leq 16\%$. Right: strong separation, inter-centroid distance constrained to $\geq 5$, k*means is 99% accurate in inferring $k$ and baselines are $\leq 57\%$.

Table 2: Performance predicting the number of clusters in synthetic data for varying degrees of cluster separation. k*means performs consistently the best, with near-perfect accuracy when $\delta = 5$.

| | mse | | | | | acc | | | | |
|---|---|---|---|---|---|---|---|---|---|---|
| | k*means | dbscan | hdbscan | xmeans | $k$-elb | k*means | dbscan | hdbscan | xmeans | $k$-elb |
| synthetic $\delta = 2$ | 306.35 | 126.10 | 414.73 | 721.54 | 266.0 | 9.00 | 4.40 | 4.00 | 3.80 | 7.80 |
| synthetic $\delta = 3$ | 81.70 | 252.41 | 116.35 | 681.97 | 296.0 | 25.40 | 5.40 | 7.80 | 16.00 | 13.40 |
| synthetic $\delta = 4$ | 1.94 | 244.28 | 28.34 | 630.13 | 274.0 | 68.00 | 7.60 | 21.40 | 22.20 | 20.10 |
| synthetic $\delta = 5$ | 0.00 | 238.18 | 12.83 | 623.99 | 274.0 | 99.80 | 6.60 | 57.60 | 25.40 | 20.90 |

gap between k*means and the baselines also widens under these conditions. k*means reaches near-perfect accuracy in the highly separable setting, c.f. the next highest of HDBSCAN at 58%. Appendix D shows the same experiment with variances that differs by cluster. Figure 1 contains visualisations of the predictions of k*means.

## 6.2 Labelled Datasets

We evaluate k*means on six datasets spanning multiple modalities. **MNIST** and **USPS** both consist of handwritten digit images from 0–9, **Imagenette** (Howard & Gugger, 2019) is a subset of ImageNet with ten image classes, **Speech Commands** consists of short spoken words for command recognition in 36 classes, **20 NewsGroups** is a dataset of text documents across twenty topics, and **MSRVTT** consists of video clips paired with natural language captions in 20 categories. For all datasets, we perform dimensionality reduction with UMAP (McInnes et al., 2018) (min-dist=0, n-neighbors=10). For ImageNet we first apply CLIP (Radford et al., 2021) and for 20 Newsgroups we first take features from Llama-3.1 (Touvron et al., 2023) (mean across all tokens). For MSRVTT, we first take CLIP features of both the video and text (mean across all frames and tokens). As well as tracking the predicted number of classes, we assess partition quality by comparing to the ground truth partition arising from the class labels using three metrics: clustering accuracy (ACC), adjusted rand index (ARI), and normalised mutual information (NMI), as defined, in Mahon & Lukasiewicz (2024).

As baselines for clustering with unknown $k$, we compare to the following: affinity propagation (damping factor = 0.5), mean shift (bandwidth = median of pairwise distances), DBSCAN (eps=0.5, min-samples = 5), HDBSCAN (eps=0.5, min-samples = 5), OPTICS, ($\xi = 0.05$, min-samples=5), XMeans (kmax=$\sqrt{\text{dataset size}}$), divisive hierarchical clustering (DivHier) using silhouette score as stopping criterion for splitting, and the elbow-method with $k$-means up to $k = 200$, computed using the public *kneed* library[1]. These methods are all described in Section 3 (see also Table 1). For XMeans, in the absence of any guidance

---

[1]https://pypi.org/project/kneed/

Table 3: Supervised metrics on labelled datasets. Methods below the dotted line are given $k$.

| | | ACC | ARI | NMI | $k$ | Num Outliers | Runtime (s) |
|---|---|---|---|---|---|---|---|
| MNIST domain = images n classes = 10 | Quickshift | 71.38 | 67.62 (0.0009) | 79.08 (0.0007) | 154.20 (0.4472) | 0.00 | 843.18 (7.0129) |
| | DBSCAN | 68.75 | 54.84 | 77.66 | 6.00 | 1 | 2.95 |
| | HDBSCAN | 79.73 | 84.84 | 70.70 | 1214.00 | 11190 | 46.24 (0.79) |
| | xmeans | 40.89 (4.75) | 34.77 (4.70) | 64.74 (2.25) | 118.80 (23.59) | 0 | 16.40 (6.41) |
| | DPC | 31.88 | 19.82 | 62.84 | 102.00 | 0.00 | 58.31 (1.8287) |
| | elbow | 87.24 (7.0176) | 80.06 (8.3068) | 85.37 (2.9301) | 10.60 (1.1738) | 0.00 (0.0000) | 210.18 (12.0703) |
| | DPMM | 55.37 (0.4706) | 44.97 (0.5174) | 73.29 (0.1056) | 35.20 (0.6325) | 0.00 (0.0000) | 53.99 (0.7465) |
| | DivHier | -1.00 | -1.00 | -1.00 | -1.00 | 0 | 36000+ |
| | k*means | **91.26 (3.56)** | **84.99 (2.97)** | **87.44 (1.14)** | 10.90 (0.32) | 0 | 3.38 (0.39) |
| | kmeans | 84.12 (8.13) | 79.64 (6.92) | 85.31 (2.73) | 10.00 | 0 | 0.09 (0.05) |
| | GMM | 86.29 (7.05) | 82.61 (6.89) | 87.41 (2.74) | 10.00 | 0 | 0.78 (0.26) |
| USPS domain = images n classes = 10 | Quickshift | 88.30 | 78.49 | 86.20 | 14.00 | 0.00 | 31.19 (1.6408) |
| | DBSCAN | 80.46 | 71.00 | 83.51 | 7.00 | 0 | 0.13 |
| | HDBSCAN | 77.49 | 82.16 | 79.09 | 108.00 | 829 | 0.80 (0.01) |
| | xmeans | 55.12 (5.03) | 46.09 (4.23) | 73.27 (1.71) | 41.00 (8.12) | 0 | 1.00 (0.43) |
| | DPC | 40.35 | 24.62 | 65.70 | 78.00 | 0.00 | 1.18 (0.0058) |
| | elbow | 79.81 (7.8221) | 71.20 (8.8348) | 82.96 (3.8259) | 7.00 (0.8165) | 0.00 (0.0000) | 11.43 (0.4058) |
| | CRP | 19.58 (0.4476) | 6.54 (0.7680) | 15.94 (1.3542) | 13.60 (1.7127) | 0.00 (0.0000) | 298.26 (41.8577) |
| | DPMM | 88.59 (0.0594) | 79.12 (0.0251) | 86.76 (0.0182) | 12.00 (0.0000) | 0.00 (0.0000) | 2.68 (0.3649) |
| | DivHier | 88.11 (0.0000) | 80.27 (0.0000) | 86.13 (0.0000) | 8.00 (0.0000) | 0.00 (0.0000) | 31.79 (2.3307) |
| | k*means | **88.68 (0.00)** | **81.57 (0.00)** | **87.14 (0.00)** | 8.00 (0.00) | 0 | 0.80 (0.26) |
| | kmeans | 79.72 (8.15) | 78.68 (6.66) | 86.41 (2.12) | 10.00 | 0 | 0.03 (0.03) |
| | GMM | 81.72 (6.76) | 80.27 (5.68) | 86.84 (1.82) | 10.00 | 0 | 0.11 (0.01) |
| Imagenet (subset) domain = images n classes = 10 | Quickshift | 56.35 | 39.18 | 60.94 | 64.00 | 0.00 | 164.37 |
| | DBSCAN | 26.09 | 3.70 | 22.00 | 3.00 | 1 | 0.22 |
| | HDBSCAN | 51.62 | 46.01 | 55.52 | 402.00 | 4193 | 1.09 (0.02) |
| | xmeans | 39.21 (3.19) | 25.53 (3.03) | 55.92 (0.88) | 70.00 (8.54) | 0 | 2.76 (0.68) |
| | DPC | 39.76 | 24.10 | 56.30 | 89.00 | 0.00 | 2.32 (0.0097) |
| | elbow | 70.14 (3.9311) | 51.53 (3.4313) | 62.68 (2.1283) | 7.90 (0.8756) | 0.00 (0.0000) | 23.85 (7.1662) |
| | CRP | 11.27 (0.1030) | 0.00 (0.0072) | 0.16 (0.0294) | 12.40 (1.9551) | 0.00 (0.0000) | 397.00 (47.4286) |
| | DPMM | 70.37 (1.4710) | 55.45 (1.3848) | 64.07 (0.9699) | 15.20 (0.7888) | 0.00 (0.0000) | 40.71 (1.3952) |
| | DivHier | 58.08 (0.0000) | 38.26 (0.0000) | 59.82 (0.0000) | 5.00 (0.0000) | 0.00 (0.0000) | 317.69 (10.5432) |
| | k*means | *66.18 (1.55)* | *46.42 (1.45)* | *60.20 (0.86)* | 6.40 (0.70) | 0 | 0.94 (0.34) |
| | kmeans | **69.79 (5.18)** | **55.08 (4.65)** | **64.16 (2.81)** | 10.00 | 0 | 0.05 (0.04) |
| | GMM | 66.85 (6.11) | 53.97 (5.44) | 64.01 (2.76) | 10.00 | 0 | 0.30 (0.09) |
| Speech Commands domain = audio n classes = 36 | Quickshift | 63.15 | 33.27 | 71.43 | 345.00 | 0.00 | 195.24 |
| | DBSCAN | 50.60 | 10.52 | 61.59 | 20.00 | 0 | 2.22 |
| | HDBSCAN | 65.35 | 67.68 | 67.12 | 2453.00 | 24170 | 53.98 (7.86) |
| | xmeans | 26.32 (7.78) | 14.33 (8.22) | 47.70 (18.56) | 190.10 (161.25) | 0 | 16.00 (13.01) |
| | DPC | 55.79 | 39.88 | 67.80 | 114.00 | 0.00 | 65.47 |
| | elbow | 62.59 (8.0219) | 40.20 (8.7028) | 66.34 (4.6024) | 21.00 (4.5704) | 0.00 (0.0000) | 395.99 (25.0698) |
| | DPMM | 62.64 (0.3104) | 46.56 (0.4448) | 70.13 (0.0886) | 66.90 (0.7379) | 0.00 (0.0000) | 66.11 (0.3408) |
| | CRP | 11.53 (0.8678) | 4.13 (0.6360) | 12.86 (0.2315) | 30.30 (3.0569) | 0.00 (0.0000) | 6991.60 (871.0678) |
| | DivHier | -1.00 | -1.00 | -1.00 | -1.00 | 0 | 36000+ |
| | k*means | *68.73 (1.57)* | *48.43 (2.49)* | *70.22 (0.67)* | 26.50 (0.97) | 0 | 20.98 (5.22) |
| | kmeans | **71.08 (1.72)** | **57.78 (1.67)** | 72.67 (0.47) | 36.00 | 0 | 0.30 (0.06) |
| | GMM | 71.04 (1.27) | 56.12 (1.63) | **72.90 (0.42)** | 36.00 | 0 | 6.46 (0.89) |
| 20 NG domain = text n classes = 20 | Quickshift | 36.62 | 21.66 | 46.05 | 218.00 | 0.00 | 339.72 |
| | DBSCAN | 16.40 | 1.98 | 18.59 | 12.00 | 0 | 0.40 |
| | HDBSCAN | 30.08 | 24.05 | 47.72 | 664.00 | 6153 | 3.27 (0.03) |
| | xmeans | 30.01 (10.66) | 15.48 (8.18) | 37.78 (19.83) | 107.60 (56.16) | 0 | 4.78 (2.51) |
| | DPC | 39.47 | 21.83 | 47.41 | 143.00 | 0.00 | 4.58 (0.1567) |
| | elbow | 40.45 (7.1649) | 23.53 (7.6900) | 43.66 (5.8133) | 10.67 (4.1633) | 0.00 (0.0000) | 99.08 (44.8268) |
| | DPMM | 49.75 (0.5689) | 31.17 (0.8372) | 50.63 (0.1245) | 45.00 (1.0000) | 0.00 (0.0000) | 36.14 (0.7786) |
| | CRP | 7.90 (0.0968) | -0.00 (0.0067) | 0.19 (0.0252) | 12.00 (2.0000) | 0.00 (0.0000) | 1657.05 (233.9947) |
| | DivHier | 18.02 (0.0000) | 5.55 (0.0000) | 20.03 (0.0000) | 2.00 (0.0000) | 0.00 (0.0000) | 6.90 (0.6898) |
| | k*means | *42.33 (1.14)* | *26.08 (0.44)* | 46.61 (0.67) | 11.20 (0.42) | 0 | 2.46 (0.96) |
| | kmeans | 46.73 (1.47) | 33.68 (0.53) | 50.42 (0.48) | 20.00 | 0 | 0.07 (0.06) |
| | GMM | **47.03 (1.22)** | **33.71 (0.78)** | **50.68 (0.50)** | 20.00 | 0 | 0.86 (0.19) |
| MSRVTT domain = video & text n classes = 20 | Quickshift | 38.80 | 14.25 | 40.68 | 236.00 | 0.00 | 99.96 |
| | DBSCAN | 37.65 | 11.51 | 39.23 | 27.00 | 0.00 | 0.05 |
| | HDBSCAN | 18.40 | 5.90 | 45.39 | 321.00 | 1275.00 | 0.26 (0.73) |
| | XMEANS | 40.64 | 21.75 | 45.13 | 78.00 | 0.00 | 0.38 |
| | DPC | 29.84 | 13.36 | 46.53 | 158.00 | 0.00 | 0.52 (0.0218) |
| | elbow | 44.44 (1.7576) | 20.26 (1.7683) | 37.74 (1.5068) | 12.20 (1.2293) | 0.00 (0.0000) | 7.65 (0.3514) |
| | DPMM | 45.51 (0.8336) | 26.30 (1.1535) | 44.91 (0.8652) | 25.60 (1.0750) | 0.00 (0.0000) | 8.48 (1.5417) |
| | CRP | 10.66 (1.2531) | 0.61 (0.2564) | 3.07 (0.1236) | 18.00 (3.4641) | 0.00 (0.0000) | 582.05 (73.6062) |
| | DivHier | 27.14 (0.0000) | 3.28 (0.0000) | 13.57 (0.0000) | 2.00 (0.0000) | 0.00 (0.0000) | 1.12 (0.2435) |
| | k*means | **44.10 (136.25)** | 25.75 (65.28) | 38.16 (33.06) | 18.10 (87.56) | 0.00 | 2.57 (40.59) |
| | kmeans | 40.07 (108.95) | 25.35 (128.11) | 38.43 (62.75) | 20.00 | 0.00 | 0.04 (1.01) |
| | GMM | 41.41 (193.57) | 25.28 (101.87) | 38.44 (49.71) | 20.00 | 0.00 | 0.31 (9.16) |

Table 4: Comparison of `k*means` with the common approach of sweeping $k$ and selecting with BIC. `k*means` is consistently faster and more accurate.

| | | ACC | ARI | NMI | NC | Runtime (s) |
|---|---|---|---|---|---|---|
| MNIST | $k$-means BIC sweep | 12.86 | 8.17 | 56.01 | 25.00 | 148.15 |
| | k*means (exhaustive) | 89.45 (5.2504) | 82.65 (4.4397) | 86.68 (1.4187) | 10.20 (0.8367) | 102.91 (3.7840) |
| | k*means | **91.26 (3.56)** | **84.99 (2.97)** | **87.44 (1.14)** | 10.90 (0.32) | **3.38 (0.39)** |
| USPS | $k$-means BIC sweep | 32.36 (0.81) | 21.77 (0.77) | 65.20 (0.40) | 68.40 (3.10) | 11.21 (0.73) |
| | k*means (exhaustive) | 86.80 (6.6134) | 79.98 (6.9232) | 86.39 (2.5467) | 8.20 (0.4472) | 4.37 (0.3921) |
| | k*means | **88.68 (0.00)** | **81.57 (0.00)** | **87.14 (0.00)** | 8.00 (0.00) | **0.80 (0.26)** |
| ImageNet (subset) | $k$-means BIC sweep | 8.16 (0.26) | 1.18 (0.05) | 7.62 (0.05) | 83.20 (4.13) | 19.54 (0.23) |
| | k*means (exhaustive) | 70.80 (4.2809) | 54.74 (2.5507) | 64.97 (1.6734) | 9.60 (0.5477) | 8.25 (1.0609) |
| | k*means | **66.18 (1.55)** | **46.42 (1.45)** | **60.20 (0.86)** | 6.40 (0.70) | **0.94 (0.34)** |
| Speech Commands | $k$-means BIC sweep | 32.19 (1.27) | 20.10 (0.90) | 62.29 (0.30) | 239.50 (12.12) | 951.58 (11.31) |
| | k*means (exhaustive) | 70.79 (0.8130) | 51.28 (1.5195) | 70.82 (0.3861) | 27.60 (1.1402) | 194.34 (6.5442) |
| | k*means | **68.73 (1.57)** | **48.43 (2.49)** | **70.22 (0.67)** | 26.50 (0.97) | **20.98 (5.22)** |
| 20 NG | $k$-means BIC sweep | 32.75 (0.54) | 17.44 (0.54) | 46.84 (0.17) | 107.30 (5.83) | 36.61 (1.28) |
| | k*means (exhaustive) | 48.02 (1.2703) | 29.46 (1.0942) | 47.47 (0.8803) | 14.60 (1.9494) | 12.08 (0.9215) |
| | k*means | **42.33 (1.14)** | **26.08 (0.44)** | 46.61 (0.67) | 11.20 (0.42) | **2.46 (0.96)** |
| MSRVTT | $k$-means BIC sweep | 27.50 (89.64) | 12.24 (53.57) | **41.36 (18.64)** | 91.60 (464.76) | 7.33 (20.57) |
| | k*means (exhaustive) | 44.62 (0.9815) | 25.82 (0.9902) | 43.85 (0.2867) | 25.40 (1.3416) | 2.05 (0.3573) |
| | k*means | **44.10 (136.25)** | **25.75 (65.28)** | 38.16 (33.06) | 18.10 (87.56) | **2.57 (40.59)** |

on selecting `kmax`, we set it to the value at which the information content is roughly equal between the index cost and the residual cost. All other parameter values are the sci-kit learn[2] defaults.

Our results are summarised in Table 3. `k*means` consistently outperforms all other methods that do not require setting $k$. Meanshift and DBSCAN tend to underestimate $k$, while affinity propagation, HDBSCAN, XMeans, and OPTICS tend to overestimate it, often by a factor of 10 or more. `k*means`, on average, slightly underestimates $k$, but is much closer than existing methods. It is also much more accurate with respect to clustering metrics, on some datasets (MNIST, USPS), even performing on par with $k$-means and GMM, which have the *true* value of $k$ specified.

Occasionally (20-NG, MSRVTT), one of the existing methods gets a high NMI score. However, we observe that they also vastly overpredict $k$ in these cases, indicating that the true and predicted partitions have very different numbers of classes. This can cause NMI to give unreliable results as the entropy in the latter is then unnaturally high. For existing methods, it is quite likely that one could obtain better results by manually tuning the parameters. We find that it is possible to get almost any value of $k$ by such tuning (see Appendix A), but the focus of the present paper is on cases in which the user does not know the true value of $k$. In other words, they do not have a ground truth against which to tune these parameters; instead, they have to use the defaults. Table 3 shows that `k*means` is a much better choice in such cases.

**Comparison to Sweeping** $k$ A common approach when clustering with unknown $k$ is to train $k$-means models with multiple values of $k$, compute some external model-selection criterion, commonly the Bayesian information criterion (Schwarz, 1978) for each, and select whichever $k$ gives the lowest BIC (Wessman et al., 2012; Zhang & Li, 2014; Lancaster & Camarata, 2019; Salmanpour et al., 2022). Table 4 shows the performance of this common approach compared to `k*means`. As we simulate the scenario where $k$ is unknown, we sweep in increments of 10% up to the dataset size. Sweeping plus BIC selection tends to favour very high values of $k$, generally 4-5x the number of annotated classes. It is also 10-50x slower than `k*means`.

**Runtime Analysis** The runtimes from Table 3 already indicate the speed of `k*means` compared with existing methods. To examine this further, and in particular how it depends on dataset size, we use subsets of varying sizes from the largest dataset in Table 3: Speech Commands, which has 99,000 data points. Figure 2 shows the runtime of `k*means`, compared to the fastest baselines, on subsets of size $1,000, 2,000, \ldots, 99,000$. The fastest at all sizes is $k$-means, which remains well under 1s even for 99,000 samples. The next is DBSCAN, rising to ∼3s, then the GMM ∼5s, and `k*means` ∼8s.

---

[2]https://scikit-learn.org/stable/

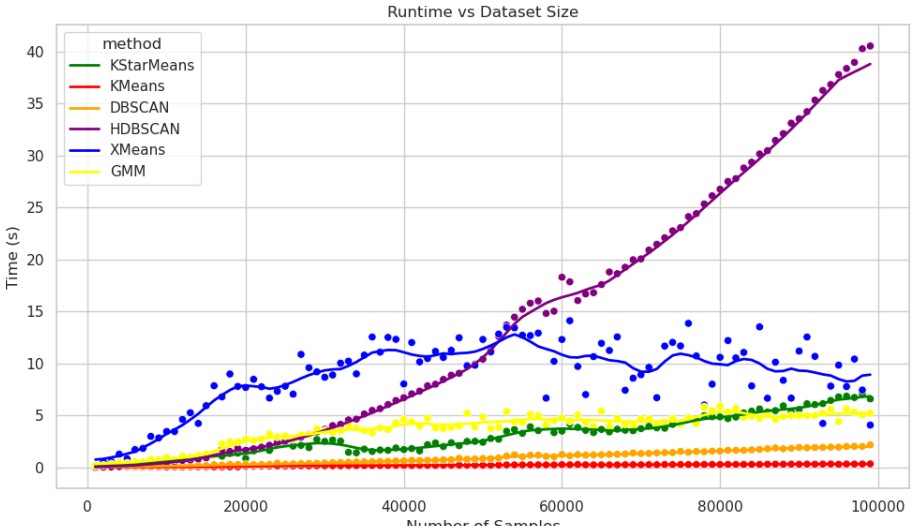

Figure 2: Windowed averages of runtime as a function of dataset size. Each point represents the mean runtime across 10 randomly sampled subsets of the given size from the Speech Commands dataset.

HDBSCAN is efficient for small samples, faster than k*means and GMM, but increases much faster, and by 99,000 samples, its runtime is 6x that of k*means. XMeans is the most erratic, by far the slowest for small sample sizes, and increases very little or even decreases, ending up close to k*means. The surprising decrease could be due to XMeans predicting fewer clusters for larger datasets. It could also be related to the optimised C-Engine that the public XMeans code makes use of[3]. Note that Figure 2 shows only the fastest five algorithms. Mean-shift, affinity propagation, and OPTICS are all substantially slower and would be off the chart if included.

## 7 Conclusion

This paper presented a new clustering algorithm, k*means, that can be applied without knowing $k$ and does not require setting any other parameters, such as thresholds. We prove that k*means is guaranteed to converge, and we show empirically on synthetic data that it can more accurately infer $k$ than comparison methods, and with near-perfect accuracy for sufficiently separated centroids. We then test it on six labelled datasets spanning image, text, audio and video domains, and show that it is significantly more accurate than existing methods in terms of standard clustering metrics. We also compare it to the standard practice of sweeping $k$ in $k$-means and selecting with a model selection criterion. Finally, we demonstrate how its runtime scales with dataset size, and show that it is faster, and scales better than the majority of existing methods. k*means can be useful in cases where the user has large uncertainty as to the appropriate value of $k$.

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

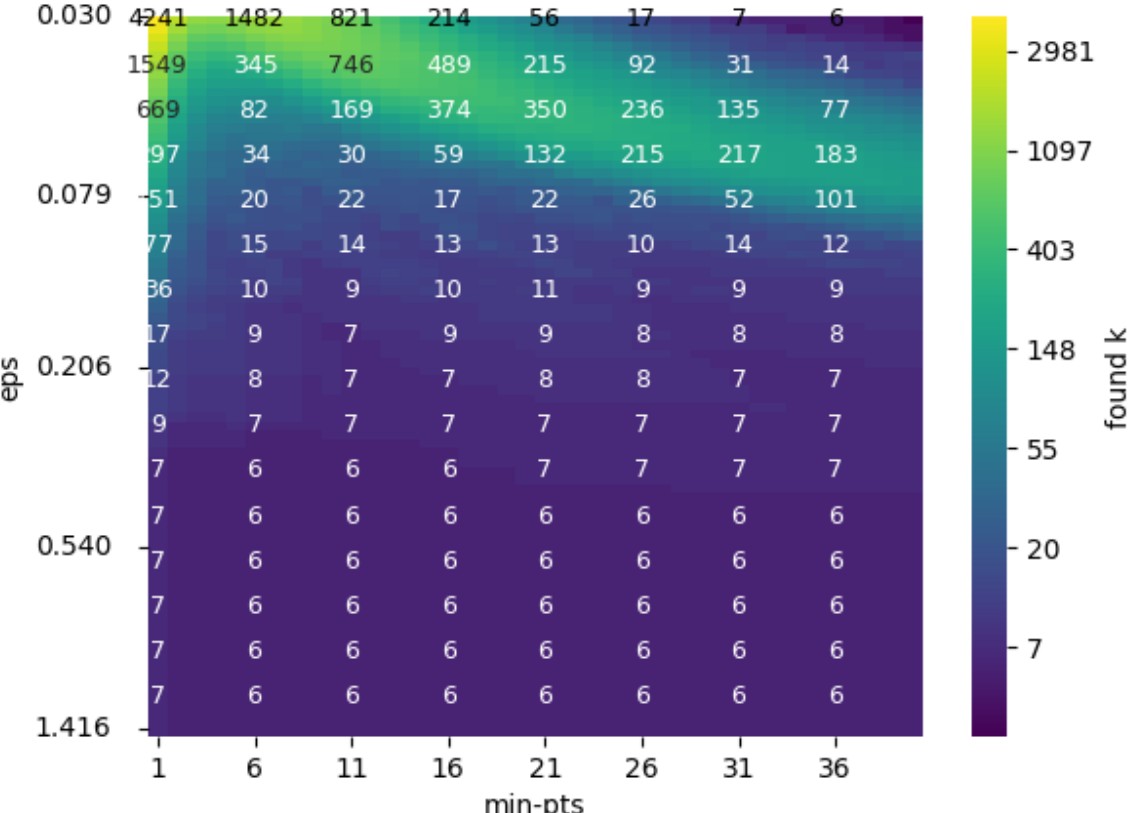

Figure 3: Values of $k$ (number of clusters) found on MNIST for different values of the DBSCAN parameters, `min-pts` (x-axis) and `eps` (y-axis). We sweep `min-pts` from 1–40, and `eps` from 0.03 to 1.5 in 5% increments.

## A Dependence of $k$ on DBSCAN Parameters

Although DBSCAN does not explicitly require setting $k$, its two key parameters, `eps` and `min-pts`, indirectly determine a value for $k$. As can be seen in Figure 3, the different values for $k$ found by DBSCAN for different values of eps and min-pts range from 6 to over 4,000. In general, smaller `eps` and smaller `min-pts` produce more clusters—the number of annotated classes is 10.

## B Derivation of Bit Cost for Clustering Objective

The objective to derive is that from Section 4.1:

$$P^* = \arg\min_{P \in \Pi(X)} |P|dm + |X|\log|P| + \frac{1}{2}\sum_{S \in P} Q(S)\,.$$

The first two terms are direct expressions of the cost to specify the centroids (each costs $dm$ bits and there are $|P|$ of them), and the cluster labels (each costs $\log|P|$ bits, and there are $|X|$ of them). The third term arises from the expression for the negative log-probability, and the fact that we can drop additive constants

in the argmin. Let $c(P, x)$ be the centroid of the cluster $x$ belongs to under partition $P$. Then

$$\operatorname*{arg\,min}_{P \in \Pi(X)} |P| dm + |X| \log |P| + \sum_{x \in X} \frac{\delta \log 2\pi + ||x - c(P, x)||^2}{2} =$$

$$\operatorname*{arg\,min}_{P \in \Pi(X)} |P| dm + |X| \log |P| + \frac{1}{2} \sum_{x \in X} ||x - c(P, x)||^2 =$$

$$\operatorname*{arg\,min}_{P \in \Pi(X)} |P| dm + |X| \log |P| + \frac{1}{2} \sum_{S \in P} \sum_{x \in S} ||x - c(P, x)||^2 =$$

$$\operatorname*{arg\,min}_{P \in \Pi(X)} |P| dm + |X| \log |P| + \frac{1}{2} \sum_{S \in P} Q(S) .$$

## C  Theoretical Guarantee of Performance for Equally-sized Spherical Multivariate Normal Clusters

Given that k*means subsumes the familiar Lloyd's algorithm for $k$-means, and given the difficulty reasoning about the behaviour of Lloyd's algorithm itself, we instead prove a guarantee of performance with respect to the $k++$ means initialisation. This is a similar approach to that taken by Ostrovsky et al. (2013).

We will prove that, if the data comes from $k$ equally-sized multivariate Normal distributions, with the same isotropic variance, separated by at least $\delta$, then the initialisation (which follows $k++$ means and selects new points in proportion to the square of their distance from previous points) produces centroids that are all with $\epsilon$ of their true values, with probability at least $a$. To simplify notation, we will assume all clusters have variance 1, but this generalises to any value as the initialisations are made with respect to relative distances and so are invariant to rescaling.

As k*means proceeds iteratively, we first analyse the single case of splitting a dataset into two, assuming it contains $k$ true clusters, where $k$ may be greater than 2. The first point is chosen randomly. The probability of it falling within $\epsilon$ of whatever cluster it is in is therefore $erf(\epsilon/\sqrt{2})$. For the second point, the probability can be expressed as a ratio. The numerator, $A$, is the integral of the squared distance from the first point times the probability density, integrated over all $\epsilon$-balls around the means of the other clusters. The denominator, $B$, is the expected value of the squared distance of a new point from the first point. We are interested in a lower bound on the probability of approximately correct cluster centroids; therefore, we consider the worst case for the location of the first point, which is that it is a distance $\epsilon$ from its centroid, and a distance $\delta - \epsilon$ from every other centroid (the latter being a lower bound via the triangle inequality). WLOG, we can assume the selected point is at the origin of $\mathbb{R}^2$, so the squared distance of a possible second point is equal to its squared norm.

Let $X \sim \mathcal{N}(\mu_x, I)$ in $\mathbb{R}^2$, where $\mu_x$ is the true centroid of $X$, and let $Z = X - \mu_x \sim \mathcal{N}(0, I)$.

We are interested in the conditional expectation:

$$\mathbb{E}[\|X\|^2 \mid \|X - \mu_x\| < \varepsilon] = \mathbb{E}[\|Z + \mu_x\|^2 \mid \|Z\| < \varepsilon] \tag{5}$$

Now expand the squared norm:

$$\|Z + \mu_x\|^2 = \|Z\|^2 + 2Z^\top \mu_x + \|\mu_x\|^2$$

Take the conditional expectation:

$$\mathbb{E}[\|Z + \mu_x\|^2 \mid \|Z\| < \varepsilon] = \mathbb{E}[\|Z\|^2 \mid \|Z\| < \varepsilon] + 2\mathbb{E}[Z^\top \mu_x \mid \|Z\| < \varepsilon] + \|\mu_x\|^2$$

The middle term vanishes, because it is an integral of an odd function about 0. The third term $||\mu_x||^2$ is lower-bounded by $(\delta - \epsilon)^2$, due to the triangle inequality and the assumption that the first sampled point is at the origin. To calculate the first term, note that $||Z||^2$ is the sum of the squares of 2 normally distributed

variables, so it has a Chi-squared distribution with 2 degrees of freedom. The $r = ||Z||^2$, then the pdf is $re^{\frac{-r^2}{2}}$. Then, we have

$$\mathbb{E}[||Z||^2 \mid ||Z|| < \varepsilon] = \frac{\int_0^\epsilon r^2 (re^{\frac{-r^2}{2}})dx}{\int_0^\epsilon re^{\frac{-r^2}{2}}dx}.$$

Substituting $u = r^2/2$, so that $du = r\,dr$ gives

$$\frac{\int_0^{\epsilon^2/2} 2ue^{-u}du}{\int_0^{\epsilon^2/2} re^{-u}}$$

The numerator becomes

$$[-(u+1)e^{-u}]_0^{\epsilon/2} = 1 + (\epsilon^2/2 - 1)e^{-\epsilon^2/2}$$

The denominator becomes

$$[-e^{-u}]_0^{\epsilon^2/2} = -e^{\epsilon^2/2} + e^0 = 1 - e^{\epsilon^2/2}.$$

So the lower bound on the conditional expectation equation 5 becomes

$$\frac{1 + (\epsilon^2/2 - 1)e^{-\epsilon^2/2}}{1 - e^{\epsilon^2/2}} + (\delta - \epsilon)^2.$$

As we are going to renormalise anyway, we instead use the unnormalised expectation

$$1 + (\epsilon^2/2 - 1)e^{-\epsilon^2/2} + (1 - e^{-\epsilon^2/2}(\delta - \epsilon)^2. \tag{6}$$

To find the probability of the new centroid being within $\epsilon$ of its true centroid, we use this total unnormalised expectation across all $k - 1$ other clusters, and normalised by the total unnormalised expectation of the squared distance. The latter contains two terms. This first is for each of the other $k - 1$ clusters, which can be computed using the same argument as above, except using the limit $\infty$ instead of $\epsilon$, giving $1 + (\delta - \epsilon)^2$. The second is for the same cluster as the first point, which can be computed in the same way except now the distance to the centroid is $\epsilon$ rather than $\delta - \epsilon$, giving $1 + \epsilon^2$. Putting this together, we get

$$\frac{(k-1)\left(1 + (\epsilon^2/2 - 1)e^{-\epsilon^2/2} + (1 - e^{-\epsilon^2/2})(\delta - \epsilon)^2\right)}{(k-1)(1 + (\delta - \epsilon)^2) + 1 + \epsilon^2} =$$

$$\frac{(k-1)\left(1 + (\epsilon^2/2 - 1)e^{-\epsilon^2/2} + (1 - e^{-\epsilon^2/2})(\delta - \epsilon)^2\right)}{(k-1)(\delta - \epsilon)^2 + k + \epsilon^2}.$$

As expected, this expression approaches 0 as $\epsilon$ approaches 0. Claim this is an increasing function of $k$. Show that the derivative wrt $k$ is always positive:

$$\frac{((k-1)(\delta-\epsilon)^2 + k + \epsilon^2)\left(1 + (\epsilon^2/2 - 1)e^{-\epsilon^2/2} + (1 - e^{-\epsilon^2/2})(\delta-\epsilon)^2\right) - (k-1)\left(1 + (\epsilon^2/2 - 1)e^{-\epsilon^2/2} + (1 - e^{-\epsilon^2/2})(\delta-\epsilon)^2\right.}{((k-1)(\delta-\epsilon)^2 + k + \epsilon^2)^2}$$

$$\Longleftrightarrow$$

$$((k-1)(\delta-\epsilon)^2 + k + \epsilon^2 - (k-1)((\delta-\epsilon)^2 + 1) > 0$$

$$\Longleftrightarrow$$

$$k + \epsilon^2 - (k - 1) > 0$$

$$\Longleftrightarrow$$

$$1 + \epsilon^2 > 0.$$

Thus, as a lower bound, we can consider $k = 2$. We want to determine what value of $\delta$ will ensure this lower bound is greater than $a$:

$$\frac{1 + (\epsilon^2/2 - 1)e^{-\epsilon^2/2} + (1 - e^{-\epsilon^2/2})(\delta - \epsilon)^2}{(\delta - \epsilon)^2 + 2 + \epsilon^2} > a$$

$$1 + (\epsilon^2/2 - 1)e^{-\epsilon^2/2} + (1 - e^{-\epsilon^2/2})(\delta - \epsilon)^2 > a(\delta - \epsilon)^2 + 2 + \epsilon^2$$

$$(1 - e^{-\epsilon^2/2})(\delta - \epsilon)^2 - a(\delta - \epsilon)^2 > 2 + \epsilon^2 - (1 + (\epsilon^2/2 - 1)e^{-\epsilon^2/2})$$

$$(\delta - \epsilon)^2 > \frac{1 + \epsilon^2 - (\epsilon^2/2 - 1)e^{-\epsilon^2/2}}{1 - e^{-\epsilon^2/2} - a}$$

$$\delta > \sqrt{\frac{1 + \epsilon^2 - (\epsilon^2/2 - 1)e^{-\epsilon^2/2}}{1 - e^{-\epsilon^2/2} - a}} + \epsilon \tag{7}$$

We want, with probability $p$, to get all initialised centroids with $\epsilon$ of their true value, which requires repeating this successfully $k$ times. The initialisations at each iteration are independent. Thus, we need

$$a^k > p$$
$$a > \sqrt[k]{p}.$$

Subbing into equation 7, we conclude that all initialised centroids will be within $\epsilon$ of their true values, with probability at least $p$, as long as

$$\delta > \sqrt{\frac{1 + \epsilon^2 - (\epsilon^2/2 - 1)e^{-\epsilon^2/2}}{1 - e^{-\epsilon^2/2} - \sqrt[k]{p}}} + \epsilon \,. \tag{8}$$

Plugging in some numbers, $p = 0.32, k = 4, \epsilon = 2.0$, we get

$$\delta > \sqrt{\frac{1 + 4 - (2 - 1)e^{-2}}{1 - e^{-2} - 0.7}} + 2$$

$$= \sqrt{\frac{4.865}{0.865 - \sqrt[4]{0.32}}} + 2.0 = 8.50 \,.$$

Thus, we conclude that, with probability at least 0.32, all centroids will be within 2 of their true values, as long as the centroids are separated by a distance of at least 8.5.

This proof assumes the parent centroid becomes one of the child centroids, but in practice it is initialised and updated via Lloyd, which would be significantly more accurate, so this is a loose bound.

## D  Extended Experimental Results

| | mse | | | | acc | | | |
|---|---|---|---|---|---|---|---|---|
| | k*means | dbscan | hdbscan | xmeans | k*means | dbscan | hdbscan | xmeans |
| synthetic s=2 | 283.44 | 43.19 | 232.47 | 735.91 | 9.60 | 7.00 | 2.60 | 2.40 |
| synthetic s=3 | 65.11 | 41.11 | 144.05 | 703.73 | 32.60 | 3.60 | 4.80 | 8.80 |
| synthetic s=4 | 1.83 | 27.12 | 104.96 | 669.22 | 71.20 | 5.40 | 12.00 | 17.00 |
| synthetic s=5 | 0.19 | 24.58 | 76.98 | 644.44 | 82.40 | 4.80 | 22.40 | 19.80 |

Table 5: Clustering performance on synthetic data where the variance differs by cluster. Variance for each cluster is sampled from a Normal distribution with mean 1 and variance 1 (thresholded at 1e-4 to prevent negative values).

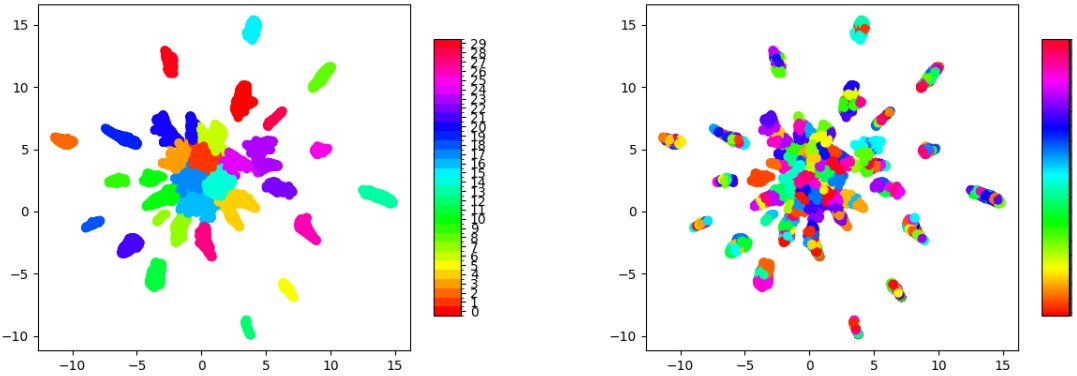

Figure 4: Clusters predicted by k*means for the UMAP representations on the Speech Commands dataset, by k*means (left) and XMeans (right). k*means predicts 33 classes and XMeans predicts 315, vs. 36 in the annotations.

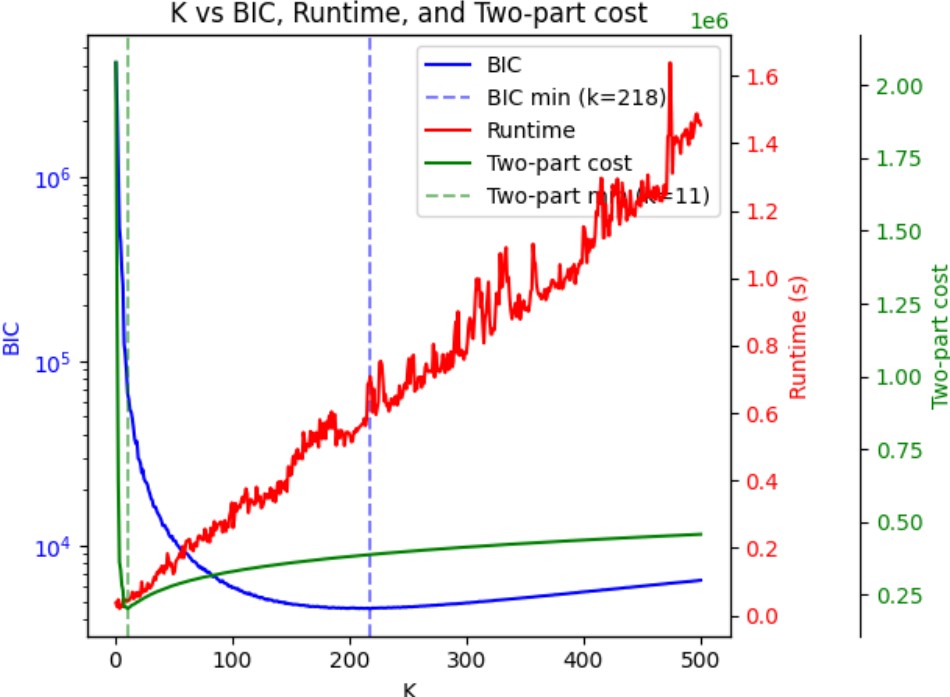

Figure 5: Different costs, as a function of $k$, for fit $k$-means models fit on the MNIST dataset. The two costs are the Bayesian information criterion (BIC) and our proposed cost from Section 4.1. The vertical dotted lines show the argmin for each cost type, which corresponds to the selected model as shown in Table 4. We also show the runtime of the $k$ means model, which is roughly linear in $k$.

