# OpenReview forum: "k∗means: A Parameter-free Clustering Algorithm"
_TMLR — Rejected by TMLR_

### Review · Reviewer_adPv · 2026-02-25

**Summary Of Contributions:**

### Summary of Contributions

The paper proposes k*means, a variant of k-means algorithm, which doesn't require predefining or tuning the number of clusters $k$, nor any other parameter. This is achieved by adopting the minimum description length (MDL) principle, trying to represent the dataset with the fewest number of bits possible. This guides the model to make decisions about splitting or merging clusters. Experiments are conducted on both synthetic and real datasets to demonstrate the effectiveness of the model in automatically determining the number of clusters, $k$, and clustering the dataset.

### Strengths

• The targeted problem of determining number of clusters, $k$, is a crucial pain point of existing algorithms.

• The paper discusses the proposed approach from thorough aspects, such as proof of convergence, time complexity, and analysis on both synthetic and real datasets.

### Weaknesses

• The proposed function to be minimized is questionable as it could behave unreasonably in some scenarios.

The optimal clustering result, or optimal partition, is supposed to be invariant to the scale of coordinate values. For instance, if the unit of the coordinates are changed and all the datapoint values are multiplied by a certain constant, the desired clustering result should be identical, since the distribution of the datapoints are not affected at all. However, In equation (1), each of the 3 terms in $P^*$ reacts to scale change differently, with the second term $|X|log|P|$ even being completely independent from the coordinate scales. This means the relative weight, or importance, of these 3 terms change as the coordinate scales change, unreasonably leading to different optimal clustering results. Similar problem occurs with dimension $d$ as well.

• The function to be minimized needs more justification and clarification to be clear.

For example, it would be great to provide more details about how Kraft-McMillan inequality leads to $− log(p(x|c))$ when representing the residual, and how a probability distribution $p(x|c)$ is needed rather than storing $||x − c||$ directly.

• The calculation of time complexity ignores some critical variables, such as number of clusters $k$.

In time complexity analysis in section 4.5, the number of clusters at the step, which is a changing variable $k$ in the context of this work, is somehow totally out of the picture. It doesn't seem to be either constant or negligible throughout the algorithm. For example, it's claimed that centroid assignment takes $O(n)$, while it actually takes at least $O(nk)$ to calculate the distance between every datapoint-centroid pair. Another example is that in the merging step, getting the closest pair of centroids is assumed to be $O(1)$, but it actually takes at least $O(k^2)$ to find the pair.

**Audience:**

Yes

**Audience Explanation:**

If the problems in the work are addressed and the conclusions still hold, a parameter free clustering algorithm is conceivably interesting to the community.

**Claims And Evidence:**

No

**Claims Explanation:**

As discussed in the "Weakness" section above.

**Requested Changes:**

1. Provide more justification and explanation about the function to be minimized.
2. Include all critical variables in the time complexity analysis, such as the number of clusters at the step (conceptually a dynamic $k$), or even dimension $d$.

---

> ### Author Response · Authors · 2026-03-16
>
> Thank you for your review and comments. Here are our responses.
>
> **Dependence on scale**
>
> Yes, the method is dependent on the scale the data is represented. Multiplying all of the features by a large number would increase the number of clusters found by k^*-means. We agree this should be made more explicit in the paper. We would in fact argue it is a feature and not a bug, and it arises from the information-theoretic basis of the method using absolute similarity scores rather than relative similarity scores. For example, consider the following two cases of clustering texts, (1) the texts are 50% news stories about politics and 50% news stories about economics, (2) the texts are 25% news stories about politics, 25% news stories about economics, and 50% excerpts from novels. Case (1) seems like there should be two clusters, one for politics and one for economics. But then consider case (2). If there should be two clusters, one for news stories and one for novel excerpts, then this would be lumping together points that in case (1) were separated. That is reasonable when $k$ is fixed, because then you are forced to choose between either distinguishing novels from news stories, or distinguishing different types of new stories from each other, and clearly the former is a more significant distinction. But when you are free to choose any value for $k$, it is not clear why points that are clustered separately in case (1) should be clustered together in case (2), given they have the same similarity score. On the other hand, you may say case (2) should have three clusters, one politics news, one for economics news and one for novels. If so, then you could imagine shrinking the scale so all points become closer to each other until the distance between novels and news stories is the same as the distance between politics and economics news stories in case (1), and the distance between politics and economic stories is the same as the distance between different types of politics stories in case (1), say those about foreign or domestic affairs. Now you would have to find two clusters, as the distances are all the same as case (1), but that would mean the shrinking had changed $k$.
>
> Essentially, we argue that it is not possible for a method to be both scale-free and capable of automatically finding $k$. It needs some information on which to base the decision, "do points X and Y belong in the same cluster?". With fixed $k$, the question instead is, "given I can only make so many cluster distinctions, which are most significant?", but in the non-fixed $k$ regime, this method of distinction cannot be employed.
>
> We have added a paragraph to the end of Section 4.1 stating that the results are not scale-free.
>
> **The function to be minimized needs more justification.**
>
> The Kraft-McMilan inequality establishes that the theoretically optimal code will behave like the logarithms of a probability distribution. However, we agree it may not be the clearest citation for the general idea of a duality between distributions and optimal codes. We have changed "..by the Kraft McMilan inequality.." -> "..using the correspondence between probability distributions and prefix codes..", with a citation to a standard general information theory textbook [1]
>
> [1] Cover, T. M., and Thomas, J. A. (2006). Elements of Information Theory
>
> **Missing $k$ parameter from complexity analysis.**
>
> We had been omitting $k$ from the complexity analysis, and focusing only on the input size, in the same way as omitting dimension or floating point precision. It is a fair point to include it however, given the focus in this work on the role of the $k$ variable. In that case, the steps, as defined in Section 4.5 have the following complexities:
>
> (1) assign and update the centroids--$O(nk)$
> (2) assign and update the sub-centroids--$O(nk)$
> (3) check whether to split--$O(nk)$
> (4) split and initialise sub-centroids--$O(n)$
> (5) check whether to merge--$O(nk + k^2)$
> (6) redefining merged cluster and reindexing the clusters--$O(k)$
>
> Overall, the complexity of $k^*$-means is then $O(nk +k^2)$. As $k$ cannot be greater than $n$ (and in reality $k \ll n$) this is $O(nk)$, which is the same as $k$-means. In practice, the $O(k^2)$ step is virtually instantaneous.

---

> > ### Comment · Reviewer_adPv · 2026-03-20
> > **Some ideas and suggestions**
> >
> > Thanks for the explanations and modifications. Most of the responses make sense to me.
> >
> > Regarding the scale-invariance of clustering algorithms, you mentioned the current approach is "using absolute similarity scores rather than relative similarity scores". On this point, personally I figure using relative similarity scores more reasonable than absolute similarity scores, as it more purely focuses on the distribution of the datapoints, rather than if the data is represented in meters or centimeters. Just as a simple example, the ratio between the longest and the shortest non-zero distance between all pairs of datapoints captures how widely a dataset spreads under its granularity. This ratio is also positively related (likely log-related) to the bits or spaces needed to describe the distances among the datapoints if we use the shortest non-zero distance between datapoints as the unit of the coordinates. This may help transform an absolute similarity score into a relative one. This may not necessarily be in the scope of this work, but might be some interesting directions for future efforts.

---

> > > ### Author Response · Authors · 2026-03-20
> > >
> > > Thank you for this suggestion. It seems to imply that one could use the inter-point distances to define a 'unit' in which the data is represented. There are complexities to deal with such as whether the units would remain axis-aligned and whether the ratios need to be integers, but the approach is in line with the one we present in this paper and one to explore in future work, as you suggest.

---

### Review · Reviewer_gDBn · 2026-03-01

**Summary Of Contributions:**

The paper proposes and explores an iterative dynamic cardinality version of k-means with cluster merge/split sub routines. The cost relies on a Gaussian assumption and an upper bound on the bits of indexing. Theoretical results detail convergence like k-means. Empirical results show the method picks a reasonable number of clusters that outperforms other methods.

**Additional Comments:**

The paper's core idea seems logical and interesting. However, it is presented currently with many errors and confusing parts. The baselines for experiments do not seem exhaustive. In particular, I request more recent versions of MeanShift like algorithms and exhaustive sweeps of k with k-means followed by BIC and proposed cost.

**Audience:**

Yes

**Audience Explanation:**

Self-tuned clustering methods that combine information criterion and algorithms are interesting to machine learning and data science community.

**Broader Impact Concerns:**

N/A.

**Claims And Evidence:**

No

**Claims Explanation:**

It impossible to say that the claims are well justified when the notation is unclear/inaccurate, with confounding notation, and cumbersome pseudocode. Please see comments below for more details.

**Requested Changes:**

**Confusing or inaccurate statements**

"this means k∗means only needs to run k-means to convergence once. " seems to be contradicted since splitting during the iterations means it is not simply k-means converging once. It should be that the k-means like updates combined with the split operations need only to converge once.

There must be a mistake in notation because it should be $\mu\in \mathbb{R}^{K\times m} $ for the mean vectors if the data is in $m$-dimensional Euclidean space. Likewise the sub-centroids should bye $K\times 2\times m$.

What is domain of $C$? Is it length $N$ vector with entries in $\{1,\ldots,K\}$?

The number of bits is upper bounded by $\log_2 K$ bits, is not equal to it.


The description of $-\log p(x|c)$ as an approximation is not concrete and the relationship with  Kraft-McMillan inequality is unclear as the inequalities applies to discrete distributions. Also $c$ has not really been defined. There should be index assignment such that it could be $\mu_c$ or $\mu_{f(x)}$ where $f:X\rightarrow \{1,\ldots,K\}$.

Why would unit variance model for the likelihood be appropriate?  Also the notation is conflicted with the previous page as now the dimension is $d$ (which is more sensical) but $m$ is now the precision...

To work with the Kraft-McMillan inequality couldn't $p$ be the probability mass over the finite dataset of length $N$ given the cluster assignment?

In the line that begins "This is the quantity"  is a bit ambiguous since it was a verbal description of the total cost followed by a parenthetical statement... This whole paragraph needs to be reorganized (state the cost mathematically) and clarified.  Especially because in Algorithm 1 the cost is computed in a different way from Equation (1); specifically, equation 1 doesn't have the $d\log(2\pi)$ term at all. As this term is constant it shouldn't matter, but its inclusion in one and not the other is confusing.

$\mathrm{Var} X$ should be defined as it operating on a set (possibly multiset as data points may be duplicated) of data. Is it the unbiased estimate of variance? I guess it has to be the biased estimate.

Section 4 begins with $N$ being the number of data points, but Algorithm 1 uses $n$. Likewise $K$ is stated as the number of clusters but then the text later uses $k$.

While it is stated in the text that it is initialized with a single cluster, Line 5 should be made more clear because the rest of the algorithm doesn't ever state how $C$ is updated...
The algorithm starts with "noend".... This may be an error with LaTeX

Why in the 'did not split' case is kmeans run again before the maybe-merge?

The operations max and min are across the whole arrays and should be stated.

In Algorithm 2 (again the noend) the notation is cumbersome and inaccurate. $x$ is a vector so there needs to be norms used rather than parentheses for sum of squares. This needs to be redesigned so someone can read it. If mathematical  pseudocode cannot be written by the authors, then simply include a listing of the code in the original NumPy or whatever framework is used....


Algorithm 3 has the problem with lacking norms but the notation is more readable...

At the start of page $SS(x)$ is not well defined. Is $x$ a vector ? If so then what is $x-x_i$ ?Later it is implied that $x$ is the average of data points, but this is contradictory...

What assumptions are involved in the claim that concludes 4.4? The proof is in the appendix but the assumptions should be stated formally in the main body.

The use of $d$ as interplant distance conflates with earlier usage... At first glance this would seem to be the dimensionality.

**Lacking more Recent Baselines for MeanShift**

Reference to mean shift should include more recent versions namely QuickShift (Vedaldi and Soatto, ECCV 2008; Rodriquez and Laio, Science 2014; and MeanShift++, Jan and Jiang, CVPR 2021.

I would expect comparisons of QuickShift could use different bandwidths and also do a sweep in terms of BIC.

**Major requested changes**

Conduct the sweep of $k$ for both BIC and the proposed cost (but with k-means iterations) exhaustively rather than increments of 10% of the dataset size, which seems to coarse. Note that in both cases naively parallelizable, so while it requires computation is not quite possible.  The use of the proposed cost after running k-means is a necessary ablation that shows that 1) the cost is a meaningful selection, and 2) the dynamic cardinality selection ends up with as good as the greedy.

**Minor requested edits**

 Page 1, 1st paragraph: References Liu et al. and Niu et al. should parenthetical.

On page 3, the blend of method names "X-means" and references "Li & Li (2009)" is confusing. It would be clearly if the algorithm by Li & Li (2009) was referred too.

 "Also similar is Ronen et al. (2022)" -> "Also the algorithm by Ronen et al. (2022) is similar."

"k-splits, Mohammadi et al." -> "The k-splits algorithm by Mohammadi et al."

"of this Section" -> "in this section"

"Python-style" -> "Python style"

Start of Section 4.2 and later on page 5 "Equation equation 1".

I don't find Table 1 very useful to the reader.

For Figure 1 use a colormap designed for categorical labels. Likewise for Figure 4.

Add reference for Bridson sampling.

Please fix the capitalization of venues to be consistent in the reference list. It should be title case.

**Questions**

The choice of exactly two sub clusters per cluster is never discussed. While this binary choice is reasonable and corresponds to most hierarchical clustering schemes, would other numbers yield similar algorithms?

---

> ### Author Response · Authors · 2026-03-16
>
> Thank you for your review. Here are our responses.
>
> **Statement that $k$-means runs to convergence just once should be updated to include split and merge.**
>
> We have updated as suggested.
>
>
> **Fix dimensionality for $\mu$ and sub-centroids.**
>
> Thank you, fixed.
>
> **What is the domain of $C$?**
>
> It is the set of all arrays of sets of data points. The $i$th element of $C$ is the set of points in the $i$th cluster. We modified the comment to state this.
>
> **Confusion around the use of Kraft McMilan inequality to derive $-\log{p(x|c)}$.**
>
> We have removed the reference to Kraft McMilan on this point and instead simply invoke the duality between distributions and prefix-free codes, as e.g. in Cover and Thomas (2006). We hope this addresses the comments about the use of the Kraft McMilan inequality and $-\log{p(x|c)}$.
>
> **The line that begins "This is the quantity" is a bit ambiguous and the paragraph needs re-organising.**
>
> We have re-organised this paragraph and the surrounding section.
>
> **Var(X) should be defined.**
>
> We have removed Var(X) here and instead define $Q(X)$ directly.
>
> **Inconsistent variable names between $N$ vs $n$ and $K$ vs $k$.**
>
> We have unified to use the lower-case version throughout.
>
> **Line 5 of Algorithm 1 should state that it is initialising as a single cluster.**
>
> We have modified the comment to state this.
>
> **Algorithms 1, 2, and 3 are showing ‘noend’.**
>
> Thank you, we have fixed this.
>
> **Why in the 'did not split' case is kmeans run again before the maybe-merge?**
>
> Actually, it is run again in the *did* merge case–note the negation in the conditional. However, we note there is an error in this section of the algorithm, which we have now corrected. The purpose of running the $k$-means step after a split or a merge is that each iteration always ends on a $k$-means step. This is a stylistic choice and our method runs essentially the same without it.
>
> **Use norms not parentheses in Algorithm 2, and make it more readable.**
>
> We have corrected to use norms, and also removed the minipage environment to give more space which should improve readability.
>
> **$SS(X)$ is not well-defined.**
>
> Actually, unless we are missing something, we believe this is well-defined. The argument $x$ is simply a variable that is used to show that the mean is the minimiser of the sum of squares. It is not necessary to state whether the $x_i$’s are vectors or their dimensionality because this proof holds for any Hilbert space. However, we acknowledge that the use of $x$ as a variable name is confusing alongside the $x_i$’s as the datapoints, so we have renamed $x$->$y$.
>
> **State the assumptions for the claim that concludes Section 4.4.**
>
> The assumptions are that the clusters are multivariate normal distributions with common variance, and that the ratio between that variance and the minimum inter-centroid distance is $d$ (now renamed to $\delta$, see below).
>
> **Confusing variable name $d$ for inter-centroid distance when it’s earlier used for dimensionality.**
>
> We have switched to use $\delta$ for inter-centroid distance.
>
> **Add Quickshift as a comparison method.**
>
> We have added this to Table 3.
>
> **Conduct the sweep of KMeans with BIC exhaustively rather than in increments of 10% of dataset size.**
>
> We actually did not use increments of 10% of the dataset size, but rather “increments of 10% up to the dataset size” (p10, paragraph 2), i.e. a geometric series with first term 1 and common ratio 1.1. This is a much dense sampling than increments of 10% *of* the dataset size. We acknowledge our statement may have been ambiguous here. In any case, we have rerun this experiment with exhaustive search up to the square root of the dataset size. The plot, which we have added to Appendix D, shows that our proposed cost increases very consistently as $k$ becomes high, so full exhaustive search has little chance of giving a different result..
>
> **Conduct the sweep of KMeans with proposed cost exhaustively.**
>
> We have added this to Table 4, searching up to N**0.5 as above.
>
> **The choice of exactly two sub clusters per cluster is never discussed.**
>
> We have observed that the description length cost function used to select $k$ in $k^*$-means is consistently u-shaped, as shown in the figure discussed above. Thus, if the global minimum would involve splitting a given cluster into more than 2 subclusters, then splitting into 2 subclusters will still give a reduction in cost, and so the cluster will be split. This means that the choice to only consider splitting into two clusters will not cause the splitting to terminate too early.

---

### Review · Reviewer_QSs7 · 2026-03-09

**Summary Of Contributions:**

The paper introduces a new clustering objective that has the advantage of not needing to specify the value of k unlike previous methods like k-means. An algorithm is used to optimize this objective and shown to converge. The paper demonstrates superior performance in comparison to existing algorithms over a collection of datasets. Also, a run-time analysis is done for the proposed algorithm.

**Audience:**

No

**Audience Explanation:**

-Overall, the paper might have a real contribution but I think it needs to improve the motivation behind objective (1) which is the main contribution of the paper.

**Broader Impact Concerns:**

The paper does not have Broader Impact Concerns.

**Claims And Evidence:**

No

**Claims Explanation:**

-I think the most important section in the paper is 4.1 where the optimization objective (1) is introduced. However, it is not clear to me or convincing why minimizing description length according to the proposed measures makes sense and would retrieve the true clustering.

-Also, the paper is making an assumption which is that the cluster distribution is a Gaussian of variance 1. This should be justified and is a limitation.

**Requested Changes:**

-Improve and expand on the motivation.

-Theorem 3, the algorithm converges to a local optimum, right? This is not a strong guarantee. Also, after how many steps? This would affect the complexity analysis for Section 4.5 since it’s only for one iteration and number of total iterations (steps) is not specified. I think this should be clarified.

Minor:
1-introduction section is misplaced before the abstract?

2-typo “Equation  equation 1” on page 4.

3-Section 4.3 is empty.

---

> ### Author Response · Authors · 2026-03-16
>
> Thank you for your time and comments. Here are our responses.
>
> **It is not clear why minimizing description length according to the proposed measures makes sense.**
>
> Minimising description length is a widely employed technique in statistics and machine learning. There is a significant body of evidence that models fit by minimising description length are useful for a variety of tasks and make accurate prediction of future data [1].  It is also closely related to maximising data likelihood, via the correspondence between code length and negative log likelihood that we make use of in our method.
>
> As to how our method minimises description length, and so utilises its effectiveness for fitting performant models, we do so by introducing a method of encoding a given dataset via a set of cluster centroids. This allows us to associate fit clustering models with encodings of the data, specifically the length of this encoding, and then search for the one with the shortest encoding length. This is laid out in Section 4.1, as you identify. However, we acknowledge that there may have been some presentation issues in this section. We have reorganised it to introduce the encoding method more generally, before describing the specific context in which we use this method, which is based on assuming a multivariate normal distribution. We hope this helps make it clear why our method minimises description length.
>
> **The paper assumes Gaussian cluster distributions with variance 1, which is a limitation and should be stated.**
>
> We acknowledge this is a limitation. We also want to state that it is common practice with MDL to work with a simple class of distributions that may not contain the true generative distribution. We have added a line to Section 4.1 acknowledging the limitation. Note also that Appendix D shows results for the synthetic experiments with differing covariance across clusters.
>
> **The complexity analysis is only for one iteration. The number of iterations is not specified.**
>
> We discuss this in Remark 4, p7: $k*$-means is guaranteed to converge, but the worst-case runtime is exponential. This is because it subsumes $k$-means for which this worst case is known. However, also like $k$-means, its empirical runtime is good, as shown by our results from Table 3 and Figure 2. Our claim is that the efficiency of $k*$-means is very similar to that of $k$-means in the following way:
> (1) each iteration is $O(N)$ (or $O(Nk)$)
> (2) it is guaranteed to converge
> (3) in the worst case, the number of iterations to converge is $O(2^N)$
> (4) in practice, it converges quickly.
> The only difference is in (4), that $k^*$-means does not converge quite as quickly as $k$-means. We would argue that these claims are adequately supported by what we have presented in the paper.
>
>
> **Typos and small fixes.**
>
> Thank you, we have fixed these.

---

### Decision · Action_Editor_ZdhH · 2026-04-21

**Recommendation:** Reject

**Audience:**

No

**Audience Explanation:**

The authors did not provide a suitable condition for the proposed algorithm, making it impractical.

**Claims And Evidence:**

No

**Claims Explanation:**

The whole review team reached the consensus that the claim made in this submission on a parameter-free clustering algorithm is not convincing.

1. Motivation. The authors argue that all existing clustering algorithms require parameters. This cannot be regarded as a drawback. Especially for the unsupervised task, each baseline has the recommended default settings.

2. Technique. The authors aim to propose a parameter-free clustering algorithm. In the proposed objective function, the coefficients of three terms in Eq. (4) can also be regarded as the parameters. Therefore, the claim made in this paper does not hold.

3. Presentation is another major issue. Reviewers have pointed out several detailed points, and I will not repeat them here.

---

> ### Author Response · Authors · 2026-04-23
>
> Thank you for your review.
>
> The Action Editor's point 2. says that the number of data points is a parameter, as well as the float precision and the dimensionality of the data. Unfortunately, this suggests a fundamental misunderstanding of what a parameter is. Not every variable in an equation is a parameter.

---

> > ### Comment · Action_Editor_ZdhH · 2026-04-23
> >
> > Dear Authors,
> >
> > The number of data points definitely is not a parameter. The coefficients of three terms in Eq. (4) are 1 vs 1 vs 1/2, which serve as the tradeoff parameters to balance the three terms. They are parameters.
> >
> > Hope the above helps.
> >
> > Kind Regards,
> >
> > AE

---

> > > ### Author Response · Authors · 2026-04-23
> > >
> > > The 1/2 term comes straight from the multivariate normal distribution, shown in Eq. (2) and (3). It is also not a parameter because it could not be anything other than 1/2. Thanks.